# *ChatGPT* identifies gender disparities in scientific peer review

**Jeroen PH Verharen***

Department of Molecular and Cell Biology and Helen Wills Neuroscience Institute, University of California, Berkeley, Berkeley, United States

**Abstract** The peer review process is a critical step in ensuring the quality of scientific research. However, its subjectivity has raised concerns. To investigate this issue, I examined over 500 publicly available peer review reports from 200 published neuroscience papers in 2022–2023. OpenAI's generative artificial intelligence *ChatGPT* was used to analyze language use in these reports, which demonstrated superior performance compared to traditional lexicon- and rule-based language models. As expected, most reviews for these published papers were seen as favorable by *ChatGPT* (89.8% of reviews), and language use was mostly polite (99.8% of reviews). However, this analysis also demonstrated high levels of variability in how each reviewer scored the same paper, indicating the presence of subjectivity in the peer review process. The results further revealed that female first authors received less polite reviews than their male peers, indicating a gender bias in reviewing. In addition, published papers with a female senior author received more favorable reviews than papers with a male senior author, for which I discuss potential causes. Together, this study highlights the potential of generative artificial intelligence in performing natural language processing of specialized scientific texts. As a proof of concept, I show that *ChatGPT* can identify areas of concern in scientific peer review, underscoring the importance of transparent peer review in studying equitability in scientific publishing.

***For correspondence:**
jeroenverharen@berkeley.edu

**Competing interest:** The author declares that no competing interests exist.

## eLife assessment

This study used *ChatGPT* to assess certain linguistic characteristics (sentiment and politeness) of 500 peer reviews for 200 neuroscience papers published in *Nature Communications*. The vast majority of reviews were polite, but papers with female first authors received less polite reviews than papers with male first authors, whereas papers with a female senior author received more favorable reviews than papers with a male senior author. Overall, the study is an **important** contribution to work on gender bias, and the evidence for the potential utility of generative AI programs like *ChatGPT* in meta-research is **solid**.

## Introduction

The peer review process is a crucial step in the publication of scientific research, where manuscripts are evaluated by independent experts in the field before being accepted for publication. This process helps ensure the quality and validity of scientific research and is a cornerstone of scientific integrity. Despite its importance, concerns have been raised regarding subjectivity in this process that may affect the fairness and accuracy of evaluations (*Park et al., 2014*; *Lipworth et al., 2011*; *King et al., 2018*; *Lee et al., 2013*; *Abramowitz et al., 1975*). Indeed, most journals engage in single-blind peer review, in which the reviewers have information about the authors of the paper, but not vice versa. While some studies have found evidence of disparities in peer review as a result of gender bias, the scope and methodology of these studies are often limited (*Blank, 1991*; *Lundine et al., 2019*). One

**eLife digest** Peer review is a vital step in ensuring the quality and accuracy of scientific research before publication. Experts assess research manuscripts, advise journal editors on publishing them, and provide authors with recommendations for improvement. But some scientists have raised concerns about potential biases and subjectivity in the peer review process. Author attributes, such as gender, reputation, or how prestigious their institution is, may subconsciously influence reviewers' scores.

Studying peer review to identify potential biases is challenging. The language reviewers use is very technical, and some of their commentary may be subjective and vary from reviewer to reviewer. The emergence of OpenAI's *ChatGPT*, which uses machine learning to process large amounts of information, may provide a new tool to analyze peer review for signs of bias.

Verharen demonstrated that *ChatGPT* can be used to analyze peer review reports and found potential indications of gender bias in scientific publishing. In the experiments, Verharen asked *ChatGPT* to analyze more than 500 reviews of 200 neuroscience studies published in the scientific journal *Nature Communications* over the past year. The experiments found no evidence that institutional reputation influenced reviews. Yet, female first authors were more likely to receive impolite comments from reviewers. Female senior authors were more likely to receive higher review scores, which may indicate they had to clear a higher bar for publication.

The experiments indicate that *ChatGPT* could be used to analyze peer review for fairness. Verharen suggests that reviewers might apply this tool to ensure their reviews are polite and accurate reflections of their opinions. Scientists or publishers might also use it for large-scale analyses of peer review in individual journals or in scientific publishing more widely. Journals might also use *ChatGPT* to assess the impact of bias-prevention interventions on review fairness.

larger study, performed by an ecology journal, found no evidence of gender bias in reviewing, but did find a bias against non-English-speaking first authors (*Fox et al., 2023*). Additionally, other factors, such as the seniority and institutional affiliation of authors, may influence the evaluation process and lead to biased assessments of research quality (*Blank, 1991*). As such, papers from more prestigious research institutions may receive better reviews (*Tomkins et al., 2017*). It is crucial to identify potential sources of disparity in the reviewing process to maintain scientific integrity and find areas for improvement within the scientific pipeline.

Natural language processing tools have shown promise in analyzing large amounts of textual data and extracting meaningful insights from evaluations (*Chowdhary, 2020*; *Hirschberg and Manning, 2015*; *Yadav and Vishwakarma, 2020*). However, applying these tools to scientific peer review has been challenging due to the specialized construction and language use in such reports. A recent study that manually annotated language use in peer reviews has shown great potential (*Ghosal et al., 2022*), but algorithms struggled to perform well in this task (*Chakraborty et al., 2020*; *Luo et al., 2021*). Recent advances in generative artificial intelligence, such as OpenAI's *ChatGPT*, offer new possibilities for studying scientific peer review. These models can process vast amounts of text and provide accurate sentiment scores and language use metrics for individual sentences and documents. As such, using generative artificial intelligence to study scientific peer review may ultimately help improve the overall quality and fairness of scientific publications and identify areas of concern in the way towards equitable academic research.

This study had three main objectives. The first aim was to test whether the latest advances in generative artificial intelligence, such as OpenAI's *ChatGPT*, can be used to analyze language use in specialized scientific texts, such as peer reviews. The second aim was to explore subjectivity in peer review by looking at consistency in favorability across reviews for the same paper. The last aim was to test whether the identity of the authors, such as institutional affiliation and gender, affect the favorability and language use of the reviews they receive.

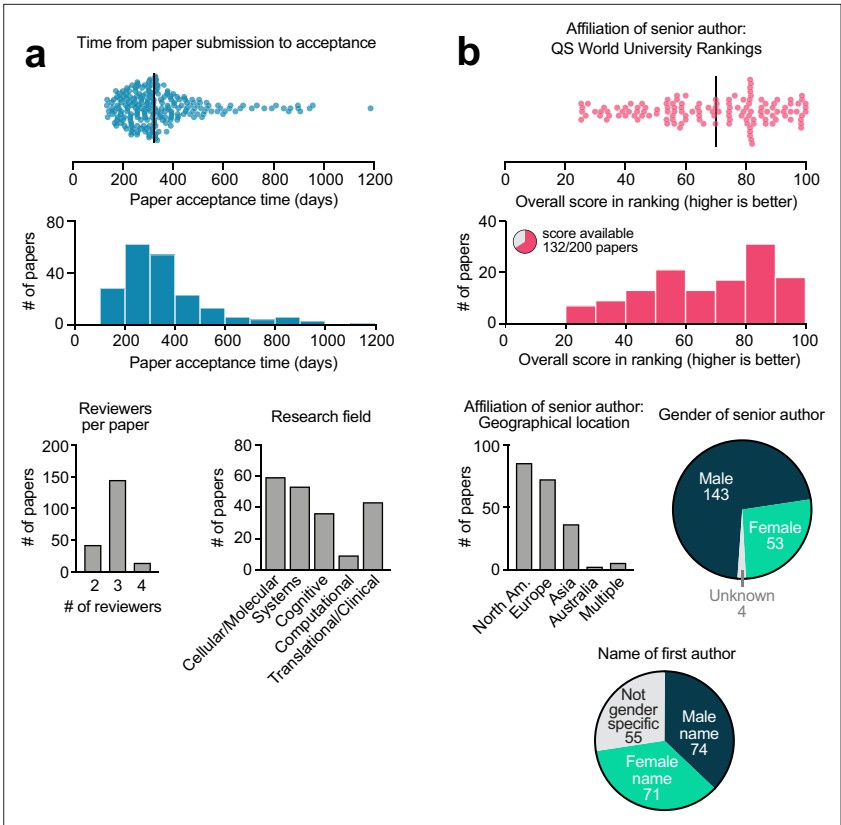

**Figure 1.** Characteristics of the 200 papers included in this analysis. (**a**) Paper metrics. (**b**) Author metrics. More information on how these metrics were collected and defined can be found in the 'Methods' section.

# Results

## An analysis of scientific peer review

*Nature Communications* has engaged in transparent peer review since 2016, giving authors the option to (and since 2022 requiring authors to) publish the peer review history of their paper (*Anonymous, 2022*). To explore language use in these reports, I downloaded the primary (i.e., first-round) reviews from the last 200 papers in the neuroscience field published in this journal. This yielded a total of 572 reviews from 200 papers, with publication dates ranging from August 2022 to February 2023. Additional metrics of these papers were manually collected (*Figure 1a and b*), including the total time until paper acceptance, the subfield of neuroscience, the geographical location and QS World Ranking score of the senior author's institutional affiliation, the gender of the senior author, and whether the first author had a male or female name (see 'Methods' for more information on classifications and a rationale for the chosen metrics). These metrics were collected to test whether they influenced the favorability and language use of the reviews that a paper received.

## Sentiment analysis

To assess the sentiment and language use of each of the peer review reports, I asked OpenAI's generative artificial intelligence *ChatGPT* to extract two scores from each of the reviews (*Figure 2a*). The first score was the *sentiment score*, and measures how favorable the review is. This metric ranges from −100 (negative) to 0 (neutral) to +100 (positive). Sentiment reflects the reviewer's opinion about the paper and is what presumably drives the decision for a paper to be accepted or rejected. The second score was the *politeness score*, which evaluates how polite a review's language is, measured on a scale from −100 (rude) to 0 (neutral) to +100 (polite). *ChatGPT* was able to extract sentiment and politeness scores for all of the 572 reviews and usually included a reasoning of how it established the score (*Figure 2—figure supplement 1*).

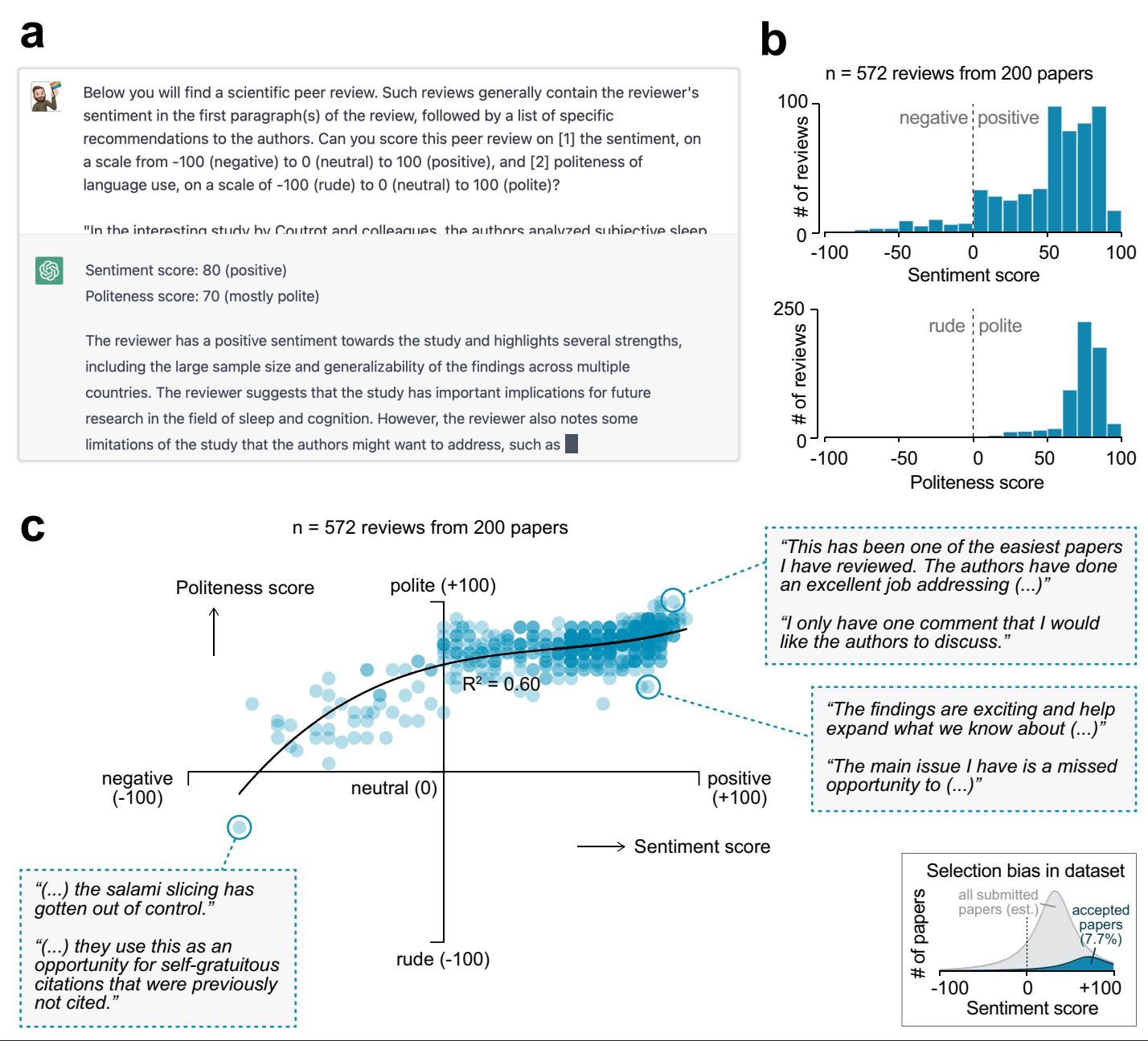

**Figure 2.** Sentiment analysis on peer review reports using generative artificial intelligence. (**a**) OpenAI's generative artificial intelligence model *ChatGPT* was used to extract a sentiment and politeness score for each of the 572 first-round reviews. Shown is an example query and *ChatGPT*'s answer. (**b**) Histograms showing the distribution in sentiment (top) and politeness (bottom) scores for all reviews. (**c**) Scatter plot showing the relation between sentiment and politeness scores for the reviews (60% variance explained in third-degree polynomial). Insets show excerpts from selected peer reviews. Inset in the bottom-right corner is a visual depiction of the expected selection bias in this dataset, as only papers accepted for publication were included in this analysis (gray area represents full pool of published and unpublished papers; not to scale).

The online version of this article includes the following figure supplement(s) for figure 2:

**Figure supplement 1.** Validation #1: examples of *ChatGPT* inputs and outputs.

**Figure supplement 2.** Validation #2: consistency in sentiment and politeness scores for two different times *ChatGPT* was asked to analyze review reports.

**Figure supplement 3.** Validation #3: example of a manually manipulated review, showing that *ChatGPT* can pick up artificial changes in sentiment and language use.

**Figure supplement 4.** Validation #4: comparison of *ChatGPT*'s scores of sentiment and politeness as compared to seven (blinded) human scorers for a diverse sample of reviews.

The accuracy and consistency of the generated scores were validated in four different ways. First, for a representative sample of the reviews, I read both the review and *ChatGPT*'s reasoning of how it came to the scores (e.g., see *Figure 2—figure supplement 1*). I established that the algorithm was able to extract the most important sentences from each of the reviews and provide a plausible score. Second, since generative artificial intelligence can provide different answers every time it is prompted, the algorithm was asked to provide scores for each review twice. This yielded a significant correlation between the first and second iterations of scoring ($p<0.0001$ for both sentiment and politeness scores; *Figure 2—figure supplement 2*); the average of the two scores was used for all subsequent analyses in this paper. Third, manipulated reviews (in which I manually re-wrote a 'neutral' review in a more rude, polite, negative, or positive manner) were input into *ChatGPT*, which confirmed that this changed the review's politeness and sentiment scores, respectively (*Figure 2—figure supplement 3*). Finally, for a subset of reviews, *ChatGPT*'s scores were compared to that of seven human scorers that were blinded to the algorithm's scores (*Figure 2—figure supplement 4*). Interestingly, there was high variability across human scorers, but their average score had a high correlation to that of *ChatGPT* (linear regression for sentiment score: $R^2 = 0.91$, $p=0.0010$; for politeness score: $R^2 = 0.70$, $p=0.018$). Importantly, *ChatGPT* was superior to the lexicon- and rule-based algorithms *TextBlob* (*Loria, 2023*) and *VADER* (*Hutto and Gilbert, 2014*) in scoring a review's sentiment; both these algorithms did not significantly predict the average human-scored sentiment (*TextBlob*: $R^2 = 0.13$, $p=0.42$; *VADER*: $R^2 = 0.07$, $p=0.56$). Together, these validations indicate that *ChatGPT* can accurately score the sentiment and politeness of scientific peer reviews and does so better than other available tools.

The majority of the 572 peer reviews (89.9%) were of positive sentiment; 7.9% were negative; 2.3% were neutral (i.e., a sentiment score of 0) (*Figure 2b*). 99.8% of reviews were deemed polite by the algorithm (i.e., a positive politeness score), only one review was scored as rude (i.e., a negative politeness score; *Figure 2c*, bottom-left inset). A regression analysis indicated a strong relation between the reviews' sentiment and politeness scores (60% of variance explained in a third-degree polynomial regression) (*Figure 2c*). Thus, the more positive a review, the more polite the reviewer's language generally is. It is important to note here that the papers included in this analysis were ultimately accepted for publication in *Nature Communications*, which has a low acceptance rate of 7.7%. As a result of this selection, there will be an over-representation of positive scores in this analysis (*Figure 2c*, bottom-right inset).

## Consistency across reviewers

If a research paper meets certain objective standards of quality, one can reasonably expect that reviewers evaluating that paper would share a common view on its overall sentiment. To investigate if this is the case, I analyzed the consistency across review scores for the same paper (*Figure 3*). As expected, the overall distribution of sentiment and politeness scores did not differ between the first three reviewers (*Figure 3a*). Interestingly, a linear regression analysis of sentiment scores across reviewers indicated very low, if any, correlation between the sentiment scores of reviews for the same paper (*Figure 3b*). The maximum variance explained in sentiment scores between reviewers was 5.5% (between reviewers 1 and 3; the only comparison that reached statistical significance). I also calculated the intra-class correlation coefficient (*Liljequist et al., 2019*) between the different reviewers, which demonstrated poor inter-reviewer reliability of scoring (ICC = 0.055, 95% confidence interval of –0.025–0.144). These results indicate high levels of disagreement between the reviewers' favorability of a paper, suggesting that the peer review process is subjective.

I then looked at the relation between a paper's review scores and its acceptance time (i.e., the time from paper submission to acceptance). For this analysis, review scores were first classified as the lowest, median (only for papers with an odd number of reviewers), or highest for a paper (*Figure 3c*). A linear regression analysis indicated that the median sentiment score was the best predictor of a paper's acceptance time ($R^2 = 0.1404$, $p<0.0001$), followed by the lowest sentiment score ($R^2 = 0.0670$, $p=0.0002$) (*Figure 3c*, bottom-left panels). Interestingly, a paper's highest sentiment score did not significantly predict acceptance time ($R^2 = 0.0088$, $p=0.1874$).

## Exploring disparities in peer review

To explore potential sources of disparities in scientific publishing, I correlated the review scores, pooled across all papers, with the different paper and author metrics that were collected earlier (*Figure 1b*).

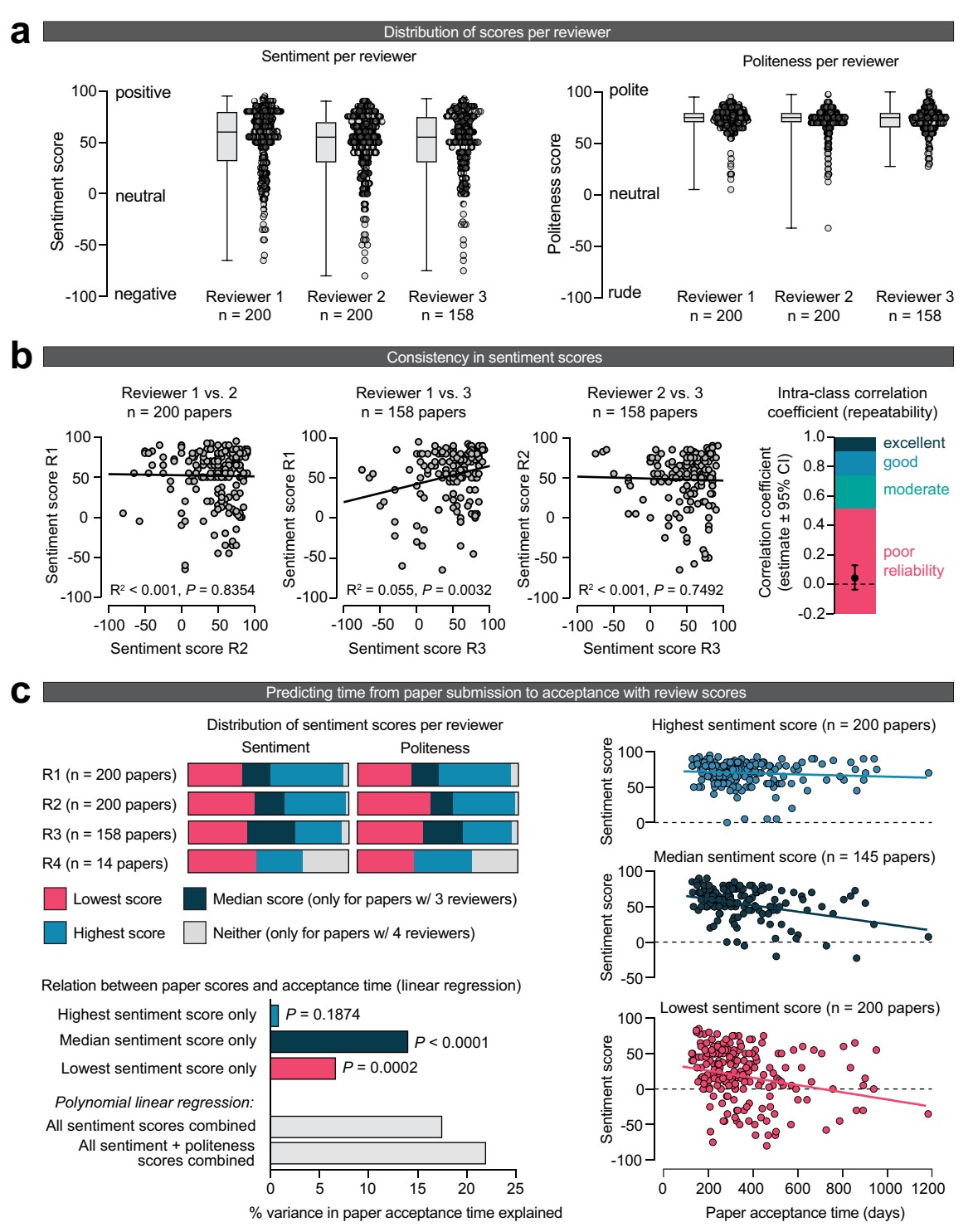

**Figure 3.** Consistency across reviews. (**a**) Sentiment (left) and politeness (right) scores for each of the three reviewers. The lower sample size for reviewer 3 is because 42 papers received only two reviews. No significant effects were observed of reviewer number on sentiment (mixed effects model, $F_{(1.929, 343.3)}$ = 1.564, p=0.2116) and politeness scores (mixed effects model, $F_{(1.862, 331.4)}$ = 1.638, p=0.1977). (**b**) Correlations showing low consistency of sentiment scores across reviews for the same paper. The sentiment scores between reviewers 1 and 3 (middle panel) is the only comparison that reached statistical significance (p=0.0032), albeit with a low amount of variance explained (5.5%). The intra-class correlation coefficient (ICC) measures how similar the review scores are for one paper, without the need to split review up into pairs. An ICC < 0.5 generally indicates poor reliability (i.e., repeatability) (*Liljequist et al., 2019*). (**c**) Linear regression indicating the relation between a paper's sentiment scores and the time between paper submission and acceptance. For this analysis, reviews were first split into a paper's lowest, median (only for papers with an odd number of reviews) and highest

*Figure 3 continued on next page*

*Figure 3 continued*

sentiment score. The lowest and median sentiment score of a paper significantly predicted a paper's acceptance time, but its highest sentiment score did not. Note that the relation between politeness scores and acceptance time was not individually tested given the high correlation between sentiment and politeness, thus having a high chance of finding spurious correlations. The metric '% variance in paper acceptance time explained' denotes the $R^2$ value of the linear regression.

No significant effects were observed between sentiment and politeness scores across the different subfields of neuroscience (*Figure 4a*). With respect to the institutional affiliation of the senior author, no effects were observed between the scores and the continent in which the senior author was based (*Figure 4b*). Additionally, no correlation was observed between the institute's score on the QS World Ranking and the paper's sentiment and politeness scores (*Figure 4c*).

Finally, I looked at how the gender of the first and senior authors may affect a paper's review scores. First authors with a female name received significantly more impolite reviews, but no effect

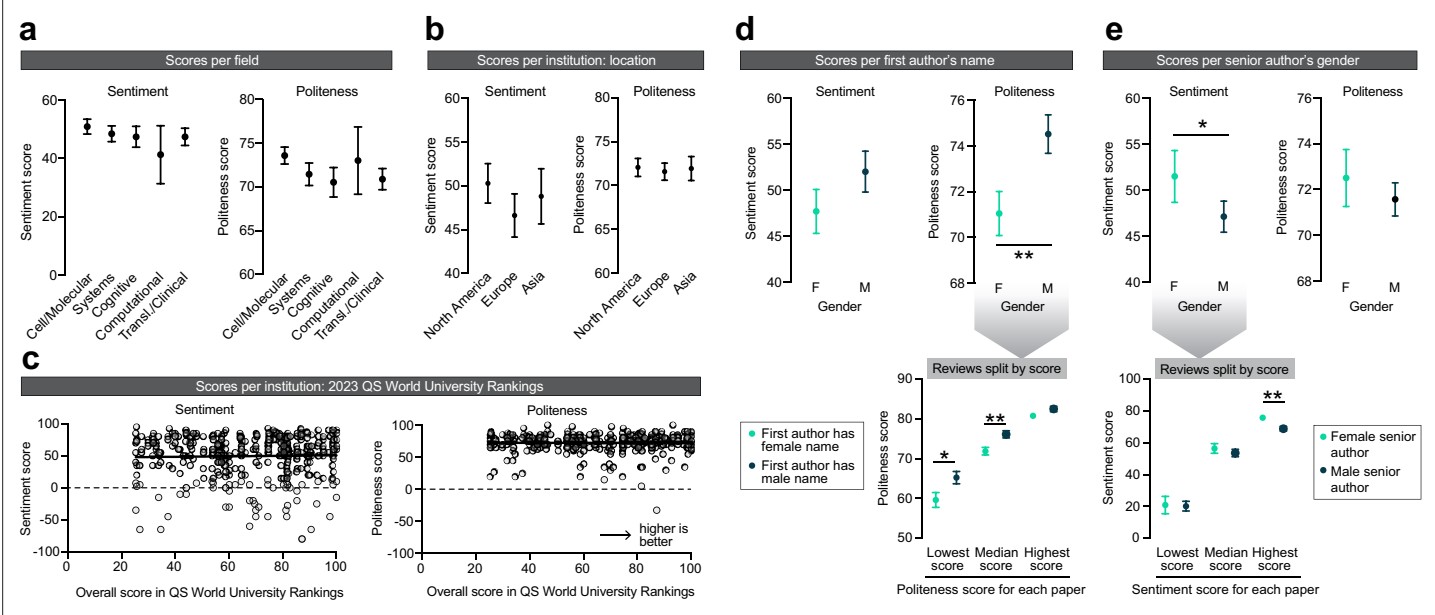

**Figure 4.** Exploring disparities in peer review. (**a**) Effects of the subfield of neuroscience on sentiment (left) and politeness (right) scores. No effects were observed on sentiment (Kruskal–Wallis ANOVA, $H = 2.380$, p=0.6663) or politeness (Kruskal–Wallis ANOVA, $H = 8.211$, p=0.0842). n = 178, 149, 100, 20, 125 reviews per subfield. (**b**) Effects of geographical location of the senior author on sentiment (left) and politeness (right) scores. No effects were observed on sentiment (Kruskal–Wallis ANOVA, $H = 1.856$, p=0.3953) or politeness (Kruskal–Wallis ANOVA, $H = 0.5890$, p=0.7449). n=239, 208, 103 reviews per continent. (**c**) Effects of QS World Ranking score of the senior author's institutional affiliation on sentiment (left) and politeness (right) scores. No effects were observed on sentiment (linear regression, $R^2 = 0.0006$, p=0.6351) or politeness (linear regression, $R^2 < 0.0001$, p=0.9804). n = 430 reviews. (**d**) Effects of the first author's name on sentiment (left) and politeness (right) scores. No effects were observed on sentiment (Mann–Whitney test, $U = 19,521$, p=0.2131) but first authors with a female name received significantly less polite reviews (Mann–Whitney test, $U = 17,862$, p=0.0080, Hodges–Lehmann difference of 2.5). Post hoc tests on the data split per lowest/median/highest politeness score indicated significantly lower politeness scores for females for the lowest (Mann–Whitney test, $U = 1987$, p=0.0103, Hodges–Lehmann difference of 5) and median (Mann–Whitney test, $U = 1983$, p=0.0093, Hodges–Lehmann difference of 2.5) scores, but not of the highest score (Mann–Whitney test, $U = 2279$, p=0.1607). n = 206 (F), 204 (M) reviews for top panels; n = 71 (F), 74 (M) papers for lower panel (but n = 54 [F], 53 [M] papers for median scores, because not all papers received three reviews). (**e**) Effects of the senior author's gender on sentiment (left) and politeness (right) scores. Women received more favorable reviews than men (Mann–Whitney test, $U = 28,007$, p=0.0481, Hodges–Lehmann difference of 5) but no effects were observed on politeness (Mann–Whitney test, $U = 29,722$, p=0.3265). Post hoc tests on the data split per lowest/median/highest sentiment score indicated no effect of gender on the lowest (Mann–Whitney test, $U = 3698$, p=0.7963) and median (Mann–Whitney test, $U = 3310$, p=0.1739) sentiment scores, but the highest sentiment score was higher for women (Mann–Whitney test, $U = 2852$, p=0.0072, Hodges–Lehmann difference of 5). n = 155 (F), 405 (M) reviews for top panels; n = 53 (F), 143 (M) papers for lower panel (but n = 39 [F], 102 [M] papers for median scores, because not all papers received three reviews). Asterisks indicate statistical significance in Mann–Whitney tests; *p<0.05, **p<0.01.

The online version of this article includes the following figure supplement(s) for figure 4:

**Figure supplement 1.** Sentiment and politeness scores for papers in different gender groups.

**Figure supplement 2.** Experiments that journals can perform to rule out gender bias in reviewing and editorial decision making.

was observed on sentiment (*Figure 4d*). To study whether these more impolite reviews for female first authors were due to an overall lower politeness score or due to one or some of the reviewers being more impolite, I split the reviews for each paper by its lowest/median/highest politeness score. I observed that the lower politeness scores for first authors with a female name were driven by significantly lower low and median scores (*Figure 4d*, bottom panel). Thus, the least polite reviews a paper received were even more impolite for papers with a female first author. Conversely, female senior authors received significantly higher sentiment scores, indicating more favorable reviews, but these reviews did not differ in terms of politeness (*Figure 4e*). An analysis of reviews split by lowest/median/highest sentiment score indicated that the reviewer who gave the most favorable review to female senior authors did so with a significantly higher score (*Figure 4e*, bottom panel). No interactions on scores were observed between the genders of the first and senior authors (*Figure 4—figure supplement 1*).

## Discussion

Peer review is a crucial component of scientific publishing. It helps ensure that research papers are of high quality and have been scrutinized by experts in the field. However, the potential for subjectivity in the peer review process has been an ongoing concern. For example, implicit or explicit bias of reviewers may lead to disparities in peer review scores on the basis of gender or institutional affiliation. In this study, I used natural language processing tools embedded in OpenAI's *ChatGPT* to analyze 572 peer review reports from 200 papers that were accepted for publication in *Nature Communications* within the past year. I found that this approach was able to provide consistent and accurate scores, matching that of human scorers. Importantly, *ChatGPT* was superior to the conventional lexicon- and rule-based algorithms *TextBlob* and *VADER* in scoring a review's sentiment. Such algorithms score a text on the basis of the frequency of certain words, and as such may have trouble analyzing scientific text with specialized constructions and vocabulary (*Ghosal et al., 2022*), as has been shown before (*Luo et al., 2021*). Altogether, the current study serves as a proof of concept for the use of generative artificial intelligence in studying scientific peer review. Such an automated language analysis of peer reviews can be used in different ways, such as after-the-fact analyses (as has been done here), providing writing support for reviewers (e.g., by implementation in the journal submission portal), or by helping editors pick the best papers or most constructive reviewers.

Notably, there are several limitations to this study. The peer review reports I analyzed are all ultimately accepted for publication in *Nature Communications*, meaning that there is a selection bias in the reviews that were included. As such, papers that have received unfavorable reviews, or papers that have not been sent out for peer review at all, were not included in this analysis. It is unclear what the gender and institutional affiliation distribution is for the papers that were ultimately unpublished. Additionally, this study only focused on the neuroscience field, and the findings may not generalize to other fields. Similarly, it is not clear if the results from this study apply to journals beyond *Nature Communications*. Future studies may expand upon this initial work by incorporating larger sample sizes and encompassing diverse scientific disciplines and journals.

Despite the said limitations, this study may reveal several key insights into the peer review process and highlight potential areas of concern within academic publishing. First, this study found that evaluations of the same manuscript varied considerably among different reviewers. This finding suggests that the peer review process may be subjective, with different reviewers having different opinions on the quality and validity of the research. Notably, some level of variability may be expected, for example, due to different backgrounds, experiences, and biases of the reviewers. In addition, *ChatGPT* may not always reliably assess a review's sentiment, adding some spurious inter-reviewer variability. That being said, the extremely low (or even absent) relation between how different reviewers scored the same paper was striking, at least to this author. This inconsistency in the evaluations emphasizes the need for greater standardization in the peer review process, with clear guidelines and protocols that can minimize such discrepancies (*Tennant and Ross-Hellauer, 2020*).

I also investigated disparities in peer review based on the institutional affiliation of the senior author of a paper. Specifically, I looked at the geographical location (continent), as well as the score of the institute in the 2023 QS World University Rankings – an imperfect metric of the institute's perceived prestige. This analysis revealed no relation of these two metrics with the sentiment and politeness of the reviews, suggesting that evaluations were not influenced by the geographical location and

perceived prestige of the senior author's research institution. This finding is encouraging and suggests that peer review may be based on the quality and merit of the research rather than the authors' research institute. That said, the identity of the peer reviewers is not known, so it cannot be tested whether reviewers have a bias with respect to authors from a more closely related country, culture, or institution (i.e., in-group favoritism). In addition, it is important to acknowledge the selection bias present in this study, in which I exclusively considered published papers. This may mask effects resulting from bias with regard to the senior author's institutional affiliation. For example, papers from less prestigious institutions may have a higher rejection rate. To address this concern, future studies could adopt a strategy such as partnering with a journal to analyze the review sentiment associated with both rejected and accepted papers.

This study further found that first authors with a female name received less polite reviews than first authors with a male name, although this did not affect the favorability of their reviews. Regardless, this disparity is worrisome as it may indicate an unconscious gender bias in review writing that may ultimately impact the confidence and motivation of (especially early-stage) female researchers. One may argue that the effect size of gender on politeness scores is small, but given the selection bias in this dataset (*Figure 2c*, bottom-right inset), this effect may be larger in the entire pool of reviewed manuscripts (i.e., rejected + accepted). To address this issue, double-blind peer review, where the authors' names are anonymized, could be implemented. Evidence suggests that this is useful in removing certain forms of bias from reviewing (*Fox et al., 2023*; *Tomkins et al., 2017*), but has thus far not been widely implemented, perhaps because some studies have cast doubt on its merits (*Alam et al., 2011*; *Snodgrass, 2006*). Additionally, reviewers could be more mindful of their language use. Indeed, even negative reviews can be written in a polite manner (*Figure 2c*), and reviewers may want to use *ChatGPT* to extract a politeness score for their review before submitting.

Additionally, female senior authors received more favorable reviews than male senior authors in this pool of accepted papers. This disparity in sentiment score in favor of women may be surprising given the wealth of data showing unconscious bias against women, including in scientific research (*Blickenstaff, 2005*; *Pell, 1996*). It is therefore likely that the observed effect is due to selection bias elsewhere in the publishing process. There may be two potential sources of this bias. The first one is that female senior authors may submit better papers to this journal than their male peers, such that the observed gender effect on sentiment is representative for the entire pool of submitted manuscripts (i.e., rejected + accepted). This could be the result of institutional barriers that lead to a small, but highly talented pool of female principal investigators (*Sheltzer and Smith, 2014*) that submits better papers than their male peers (*Hengel, 2022*). Alternatively, women may have a higher level of self-imposed quality control (*White, 2003*), such that men submit more variable quality papers to high-impact journals like *Nature Communications*. In the imperfect process that is editorial decision-making, this may lead to the publication of certain lower-quality papers from male senior authors. The second explanation may be related to an (unconscious) selection bias in the editorial process (*Matías-Guiu and García-Ramos, 2011*), requiring female senior authors to have better papers before being sent out for peer review, or better scores before being invited for a revise-and-resubmit. As such, paper acceptance may serve as a collider variable (*Holmberg and Andersen, 2022*; *Griffith et al., 2020*), inducing a spurious association between gender of the senior author and sentiment score. Further research is required to investigate the reasons behind this effect and to identify in what level of the publishing system these differences emerge. In *Figure 4—figure supplement 2*, I propose three different experiments that journals can perform to rule out bias in reviewing or the editorial process.

Together, this study serves as a proof of concept for the use of generative artificial intelligence in analyzing scientific peer review. *ChatGPT* outperformed commonly used natural language processing tools in measuring sentiment of peer reviews and provides an easy, non-technical way for people to perform language analyses on specialized scientific texts. Using this approach, areas of concern were discovered within the academic publishing system that require immediate attention. One such area is the inconsistency between the reviews of the same paper, indicating some level of subjectivity in the peer review process. Additionally, I uncovered possible gender disparity in academic publishing and reviewing. This research underscores the potential of generative artificial intelligence to evaluate and enhance scientific peer review, which may ultimately lead to a more equitable and just academic system.

## Methods

### Downloading reviews

Reviewer reports were downloaded from the website of *Nature Communications* in February 2023. Only papers that were categorized under *Biological sciences > Neuroscience* were included in this analysis. Not all papers had their primary reviewer reports published; to reach the total of 200 papers with primary review reports, the most recently published 283 papers were considered (published between August 16, 2022, and February 17, 2023).

Additional paper metrics were subsequently collected. Paper submission and acceptance date were downloaded from the '*About this article*' section on the paper website. Paper acceptance time was calculated by counting the number of days between these two dates. Research field was manually categorized on the basis of title and abstract of the paper into five different subfields. The affiliation of the senior author was downloaded from the paper website and manually categorized based on continent; if the senior author had affiliations across multiple continents, it was categorized as 'multiple' and not used for further analyses (this was the case for five papers). The affiliated institutions' score in the 2023 QS World Ranking was downloaded from the QS World Ranking website (TopUniversities. com) in March 2023; the maximum score an institution could receive was 100. Not all institutions were listed in the QS World Ranking, usually because they were not considered an organization of higher education. If a senior author had multiple affiliations, then the affiliation with the highest score was used. Name-based gender categorization of the first author was performed using *ChatGPT* (query: '*Of the following list of international full (first + last) names, can you guess, based on name only, if these people are male, female, or unknown (i.e., name is not gender specific)?*'). As a confirmation, a representative subset of names that were assigned a gender by *ChatGPT* were verified using the Genderize database (http://genderize.io; probability > 0.5). The gender of the senior author was categorized in a similar manner, except that the categorization for gender-unspecific names was manually completed, usually by looking up the senior author on the research institution's website or the author's Google Scholar or Twitter/X profile. In this manual look-up, I tried to find the senior author's preferred pronouns. If not available, I inferred the senior author's gender on the basis of a photograph. I did not find evidence that any of the senior authors included in this analysis identified as non-binary; for four senior authors, I was not able to find or infer their gender. Note that this gender look-up was performed for the senior author, but not for the first author, for two reasons. First, first authors generally had less of an online presence than seniors authors, and it was challenging to reliably assess their gender identity. Second, I presumed that reviewers are more likely to be familiar with the senior author of papers they review (e.g., through conferences) than with first authors. As such, reviewers themselves may infer the gender of the first author on name only.

### Sentiment analysis

Scores of sentiment and politeness of language use of each peer review report were performed using OpenAI's *ChatGPT* (GPT-3.5, version February 13, 2023). The prompt consisted of the following question (see *Figure 2a*):

Below you will find a scientific peer review. Such reviews generally contain the reviewer's sentiment in the first paragraph(s) of the review, followed by a list of specific recommendations to the authors. Can you score this peer review on (1) the sentiment, on a scale from –100 (negative) to 0 (neutral) to 100 (positive), and (2) politeness of language use, on a scale of –100 (rude) to 0 (neutral) to 100 (polite)?

followed by the full text of the peer review. This question was entered into *ChatGPT* twice and the average of these scores was used for further analyses; for a correlation between the two iterations, see *Figure 2—figure supplement 2*. Note that *ChatGPT* has become more reliable in recent updates, such that different iterations of scoring now produce a highly reproducible score (see *Figure 2— figure supplement 2*).

### Statistics

To test the consistency across different reviewers of the same paper (Aim 2; *Figure 3*), I used a combination of a mixed model, linear regression models, and intra-class correlation coefficients. For *Figure 3a* (differences between reviewers 1, 2, and 3), a mixed effects model was used to compute statistical significance because repeated measures data was not always available (i.e., not all papers

received a third review). This analysis was performed in Prism 9 (GraphPad Inc). Linear regression and intra-class correlational analyses in *Figure 3b* (sentiment scores across reviewers) and *Figure 3c* (review scores vs. paper acceptance time) were performed using JASP 0.16 (University of Amsterdam). For the intra-class correlational analyses of *Figure 3b*, ICC type ICC1,1 was used; because ICC is particularly sensitive to the assumption of normality, sentiment scores were first log transformed. For the polynomial linear regression in *Figure 3c*, data were centered by z-scoring the individual sentiment and politeness scores.

To test the effects of author identity on review scores (Aim 3; *Figure 4*), I used a combination of the Kruskal–Wallis ANOVA and Mann–Whitney tests. Note that review scores were not always normally distributed, so non-parametric tests were mostly used. To compute statistical significance in *Figure 4a* (scores per field) and *Figure 4b* (scores per institution location), a Kruskal–Wallis ANOVA was used. For *Figure 4c* (scores correlated with 2023 QS World University Ranking), significance was calculated using linear regression. For *Figure 4d and e*, Mann–Whitney tests were used to compute significance between male and female authors. Significant effects were further studied by splitting the reviews per score (i.e., splitting in the lowest, median, and highest scores per paper). To calculate statistical significance between male and female authors for the lowest/median/highest score in *Figure 4d and e*, Mann–Whitney tests were used. Statistical tests were always two-tailed. All analyses in *Figure 4* were performed in Prism 9 (GraphPad Inc). Significance was defined as $p<0.05$ and denoted with asterisks; $*p<0.05$, $**p<0.01$, $***p<0.001$.

## Acknowledgements

I thank Amanda Tose, Han de Jong, Stephan Lammel, Dennis Beerdsen and James Gearon for helpful comments on this manuscript.

## Additional information

### Funding
No external funding was received for this work.

### Author contributions
Jeroen PH Verharen, Conceptualization, Resources, Data curation, Formal analysis, Validation, Investigation, Visualization, Methodology, Writing – original draft, Project administration, Writing – review and editing

### Author ORCIDs
Jeroen PH Verharen  https://orcid.org/0000-0001-7582-802X

Reviewer #1 (Public Review): https://doi.org/10.7554/eLife.90230.3.sa1
Reviewer #2 (Public Review): https://doi.org/10.7554/eLife.90230.3.sa2
Author Response https://doi.org/10.7554/eLife.90230.3.sa3

## Additional files

### Supplementary files
• MDAR checklist
• Source data 1. Peer review reports included in this paper.

### Data availability
All data are available as a source data file to this paper.

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
