## [Editor Report · eLife assessment]

This study used *ChatGPT* to assess certain linguistic characteristics (sentiment and politeness) of 500 peer reviews for 200 neuroscience papers published in *Nature Communications*. The vast majority of reviews were polite, but papers with female first authors received less polite reviews than papers with male first authors, whereas papers with a female senior author received more favorable reviews than papers with a male senior author. Overall, the study is an **important** contribution to work on gender bias, and the evidence for the potential utility of generative AI programs like *ChatGPT* in meta-research is **solid**.

---

## [Referee Report · Reviewer #1 (Public Review)]

Summary:

The author uses CHAT GPT in the assessment linguistic characteristics of peer reviews published from August 2022 to February 2023 in Nature Communications in neuroscience field. The author analysed over 500 reviews, which greatly varied in terms of author characteristics, peer review length, subfield, number of reviews and writing style. Chat GPT analysed reviews and gave the scores regarding the language characteristics related to sentiment score and politeness.

Strengths:

The innovative method is the biggest strength of this article. Moreover, the method can be implemented across fields and disciplines. I myself would like to see this method implemented in a grander scale. The author invested a lot of effort in data collection and I especially commend the that the chat GPT assessed the reviews twice, to ensure greater objectivity.

Weaknesses:

The weaknesses listed in my Public Review of the previous version have been addressed in this revised version.

---

## [Referee Report · Reviewer #2 (Public Review)]

Summary

In this study, single author Jeroen Verharen investigates 500 publicly available peer review documents from 200 neuroscience papers. He uses ChatGPT to examine the sentiment and politeness of each review and performs a series of analyses including scores across reviewers, by field, institution ranking, and author gender. This is an impressive amount of analysis for a single author and uncovers an interesting pattern where female first authors receive consistently less polite reviews compared with male first authors. It is well known that women scientists face systematic discrimination across the field, and consistently in peer review. Using ChatGPT to examine these with a predefined scoring and metric system is novel and an accessible way for others in the future to evaluate these.

Strengths include:

1. Given the variability in responses from ChatGPT, he pooled two scores for each review and demonstrated significant correlation between these two iterations. He confirmed also reasonable scoring by manipulating reviews. Finally, he compared a small subset (7 papers) to human scorers and again demonstrated correlation with sentiment and politeness.

2. The figures are consistently well presented and informative. Figure 2C nicely plots the scores with example reviews. The supplementary data are also thoughtful and include combination of first/last author genders. It is interesting that first author female last author male has the lowest score.

3. A series of detailed analysis including breaking down reviews by subfield (interesting to see the wide range of reviewer sentiment/politeness scores in Computational papers), institution, and author's name and inferred gender using Genderize. The author suggests that peer review to blind the reviewers to authors' gender may be helpful to mitigating the impoliteness seen.

4. The author has strengthened the analysis in this revision by comparing it to lexicon- and rule-based algorithms TextBlob and VADER.

Weaknesses:

The weaknesses listed in my Public Review of the previous version have been adequately addressed in this revised version, and the article now acknowledges its limitations (ie, it is a pilot, proof-of-concept study, limited to articles about neuroscience). The author proposes further studies and it will be interesting to see the results of these.

---

## [Author Response]

The following is the authors’ response to the original reviews.

Summary of changes

I thank the reviewers for their thorough feedback on this paper and providing me with such a detailed list of recommendations. I have been able to incorporate many of their suggestions, which I believe has greatly improved this paper.

The most important changes:

• I added comparisons to the lexicon- and rule-based sentiment algorithms TextBlob and VADER to Supplementary Fig. 4. This shows the superiority of ChatGPT in scoring the sentiment of scientific texts compared to existing and already-validated tools for sentiment analysis based on natural language processing. [Suggestion Reviewer 2]

• I added the measure intra-class correlation to Fig. 3b, emphasizing the inconsistency in sentiment scores across different reviews of the same paper. [Suggestion Reviewer 3]

• I added Supplementary Fig. 6, in which I directly propose different experiments to test the causes of the observed gender effects on peer review. [Suggestion Reviewer 3]

• I further studied the issue of variability in responses by ChatGPT (Supplementary Fig. 2), and learned that this has greatly improved in the latest version of ChatGPT (for Version Aug 3, 2023, R2 values of 0.99 (sentiment) and 0.86 (politeness) were reached). I show these findings in Supplementary Fig. 2. [Suggestions Reviewers 1 and 3]

• Throughout the manuscript (most notably in the Abstract and Discussion), I emphasize that this is a proof-of-concept study, and make suggestions on how to scale this up across journals and fields. I also toned down certain claims given the relatively small sample size of this study, including in the abstract. I also more prominently and elaborately discuss the limitations of the study in the Discussion section. [Suggestions Reviewers 1, 2 and 3]

• I made many smaller changes to text, figures and references on the basis of the reviewers’ comments. [Suggestions Reviewers 1, 2 and 3]

Notably, Reviewer 3 has provided me with a very detailed list of recommendations for follow-up experiments. I appreciate their ideas, and I am currently considering different options for future work. Specifically I am looking to team up with a journal to perform the experiments laid out in Supplementary Fig. 6 of the new paper, to study whether I can find evidence of bias across rejected and accepted papers. As suggested by this reviewer, I am also looking into ways to automate data collection using APIs, and by utilizing the rapidly expanding databases for transparent peer review.

Based on this preprint, I have received messages from academics that are interested in using generative AI to study scientific texts. By revising this manuscript, I hope to provide them with the tools to concurrently expand the analysis of peer review into different scientific disciplines and journals.

**Reviewer #1 (Public review)**
Strengths:The innovative method is the biggest strength of this article. Moreover, the method can be implemented across fields and disciplines. I myself would like to see this method implemented in a grander scale. The author invested a lot of effort in data collection and I especially commend that ChatGPT assessed the reviews twice, to ensure greater objectivity.

I want to thank this reviewer for commending the innovative methodology of this study. I appreciate that this reviewer would like to see this methodology implemented at a grander scale, which is a view that I share. I initially only included Neuroscience papers, because I was uncertain whether I would be able to properly assess the reviews from different scientific disciplines (and thus judge whether ChatGPT was able to provide plausible scores).

The reviewers have provided me with a list of potential follow-up experiments, and I am currently considering different options for future work. Specifically I am looking to team up with a journal to perform the experiments laid out in (the new) Supplementary Fig. 6 of the new paper, to study whether I can find evidence of bias across rejected and accepted manuscript of a journal. In addition, as suggested by Reviewer #3, I am looking into ways to automate data collection using APIs, and by utilizing the rapidly expanding databases for transparent peer review. Importantly, based on this preprint, I have received messages from academics that are interested in using generative AI to study scientific texts. By revising this manuscript now, I hope to provide them with the tools to concurrently expand the analysis of peer review into different scientific disciplines and journals.

The comments I received from the different reviewers made me realize that I did not describe the intent of this paper well enough in the original submission. I rewrote much of the Abstract, to emphasize the proof-of-concept nature of this study, and rewrote the Discussion to focus more on the limitations of the study.

Weaknesses:I have several concerns regarding the methodology of the article. The first relates to the fact that the sample is not random. The selection of journal and inclusion and exclusion criteria do not contribute well to the strength of the evidence.

Indeed, the inclusion of only accepted manuscript from a single journal is the biggest caveat of this paper. I have re-written much of the Abstract to emphasize that this is a proof-of-concept paper, hoping that other researchers concurrently expand this method to larger and more diverse datasets.

An important methodological fact is that the correlation between the two assessments of peer reviews was actually lower than we would expect (around 0.72 and 0.3 for the different linguistic characteristics). If the ChatGPT gave such different scores based on two assessments, should it not be sound to do even more assessments and then take the average?

This was a great recommendation by this reviewer, and a point also raised by Reviewer #3. Based on their suggestion, I looked into how each additional iteration of scoring would reduce the variability of scoring for a subset of papers (thus being able to advice users on an optimal number of iterations).

Interestingly, I observed that ChatGPT has become significantly more reliable in providing sentiment and politeness scores in recent versions. For the latest version (ChatGPT Aug 3, 2023), R2 = 0.992 for sentiment and R2 = 0.859 for politeness were reached for two subsequent iterations of scoring. Unfortunately, OpenAI does not allow access to previous version of ChatGPT, so the current dataset could not be re-scored. Yet, based on these data, there may no longer be a need for people to perform repeated scoring. I show these data in Supplementary Fig. 2, as I believe this is very useful information for people who are interested in using this tool.

**Reviewer #1 (Recommendations to author)**
I had some difficulties reading the article, so it would maybe help to structure the article more (e.g. In the introduction there are three aims stated, so the Statistical Analysis section could be divided in three sections, and instead of the link to figures, the author could state which variables were analysed in a specific manner) to be easier to comprehend the details. Also, I found on one place that the sample consisted of 572 reviews, and on other that it was 558.

These are very good points. I re-wrote the statistical analysis for clarity (Page 7 of the manuscript). The 558 reviews was a mistake from my part, as I forgot to include the fourth review for the 14 papers that received four reviews in the histograms of Fig. 2b and the accompanying text. This has been updated.

For figures 1a and 1b it could be considered to enter the table instead of several figures.

I thank the reviewer for pointing this out. I tried this suggestion, but I found it to reduce the readability of the paper. As an alternative, I now provide an Excel spreadsheet with all the raw data, so people can find all the characteristics of the included papers.

99.8% of the reviews analysed were assessed as polite. This is, in my opinion, extremely important finding, which shows that reviewers are still holding to certain degree of standards in communication, and it can be mentioned in the abstract.

I very much agree with this reviewer; this has now been added to the Abstract.

In results you state that QS World Ranking is "imperfect" measure. When stating that in the results section, it poses the question why it is used in the study, so maybe it is more suitable for the discussion.

This point is well taken. Even though the QS World Ranking score is imperfect, I still think it can be useful, as a rough proxy of perceived prestige of an institution. I now removed this “imperfect measure” statement from the Results section, and moved it to the Discussion (Page 5).

In the Results section, instead of using only p values, please add measures of effect (correlations, mean differences), to make it easier to place in the context.

For the significant effects of Fig. 4, I have added these to the figure legends. Please note that the used statistical tests are non-parametric, so I reported the Hodges-Lehmann differences (which is the median of all possible pairwise differences between observations from the two groups).

I think the results interpretation should be softened a bit, or the limitations of the study should be placed as the second paragraph in the discussion, since this was only specific journal with specific subfield.

I agree with this reviewer that the relatively small sample size of this paper demands more careful wording. Throughout the manuscript, I have toned down claims, and emphasized the “proof of concept” nature of this study (for example in the Abstract). I also moved the limitations section to the second paragraph of the Discussion, and elaborate more on the study’s caveats.

Methods:The measure Review time was assessed from submission to acceptance, but this does not need to be review time since it takes a lot of time sometimes to find reviewers. that needs to be stated as the limitation.

This point is well taken. I changed this to “Paper acceptance time” in Fig. 3 and the accompanying text.

Gender name determination methods differed between the assessment of the first authors and the last authors, and that needs stronger explanation.

I appreciate this reviewer raising this point, which has also been raised by Reviewer #3. For this paper, I have carefully weighed the pros and cons of automated versus manual gender determination. Initially, my intention was to rely only on a programmatic method to identify authors' names. However, I came to realize that there were inaccuracies in senior author gender predictions made by ChatGPT/Genderize. This was evident to me due to my personal familiarity with some of these authors, either because they are famous or through personal interactions. It seemed problematic to me to proceed with this analysis knowing that these misclassifications would introduce unnecessary variability to the dataset.

The advantage of the relatively small sample size in this study was the opportunity to manually perform this task, rather than being fully dependent on algorithms. While I attempted manual gender identification for the first author as well, this was way more challenging due to their limited online presence. The discrepancy in gender identification accuracy between first and senior authors did not go unnoticed, and I acknowledge the issue it presents. I also recognize that, unlike senior authors, reviewers may not necessarily be familiar with the first authors of the papers they evaluate, as indicated in the original submission of this paper. In light of this, I sought input from several PIs who often serve as reviewers. Their feedback confirmed that they typically possess knowledge of senior authors' identities, for example through conferences, whereas the same is not true for first authors. Yet, this may be different for other scientific disciplines, where the pool of reviewers might be bigger.

Notably, for future studies I may make a different decision, especially when I use larger datasets that require me to automate the process.

I also realize that my rationale for the different methods of gender determination was not explained well enough in the original submission; I now explain my reasoning more elaborately on Page 7 on the manuscript.

For sentiment analysis: Please state based on what the GPT made a decision? Which program? (e.g. for gender it used genderize.io)

This has been added to Page 7.

Finally, your entire analysis can be made reproducible (since everything is publicly available). You can share ChatGPT chats as online materials with variables entered with the dataset analysed and the code. This would increase the credibility of the findings.

I will make the entire raw dataset available through the eLife website, including all reviews and their scores.

**Reviewer #2 (Public review)**
Strengths include:1. Given the variability in responses from ChatGPT, the author pooled two scores for each review and demonstrated significant correlation between these two iterations. He confirmed also reasonable scoring by manipulating reviews. Finally, he compared a small subset (7 papers) to human scorers and again demonstrated correlation with sentiment and politeness.1. The figures are consistently well presented and informative. Figure 2C nicely plots the scores with example reviews. The supplementary data are also thoughtful and include combination of first/last author genders. It is interesting that first author female last author male has the lowest score.1. A series of detailed analysis including breaking down reviews by subfield (interesting to see the wide range of reviewer sentiment/politeness scores in computational papers), institution, and author's name and inferred gender using Genderize. The author suggests that peer review to blind the reviewers to authors' gender may be helpful to mitigating the impoliteness seen.

Thank you.

Weaknesses include:1. This study does not utilize any of the wide range of Natural Language Processing (NLP) sentiment analysis tools. While the author did have a small subset reviewed by human scorers, the paper would be strengthened by examining all the reviews systematically using some of the freely available tools (for example, many resources are available through Hugging Face [https:// huggingface.co/blog/sentiment-analysis-python ]). These methods have been used in previous examinations of review text analysis (Luo et al. 2022. Quantitative Science Studies 2:1271-1295). Why use ChatGPT rather than these older validated methods? How does ChatGPT compare to these established methods? See also: colab.research.google.com/drive/1ZzEe1lqsZIwhiSv1IkMZdOtjPTSTlKwB?usp=sharing

This was a great recommendation by this reviewer, and I have tested ChatGPT against TextBlob and VADER, the two algorithms also used by the Luo et al. study — see Supplementary Fig. 4. Perhaps unsurprisingly, these algorithms performed very poorly at scoring sentiment of the reviews. Please note that I also tested these two algorithms at scoring individual sentences, Tweets and Amazon reviews, which it did very well (i.e., the software package was working correctly). Thus, ChatGPT is better at scoring scientific texts than TextBlob and VADER, likely because these algorithms struggle with finding where in the review the sentiment is conveyed. I now discuss this on Pages 1, 3 and 4 of the manuscript.

1. The author's claim in the last paragraph that his study is proof of concept for NLP to analyze peer review fails to take into account the array of literature already done in this domain. The statement in the introduction that past reports (only three citations) have been limited to small dataset sizes is untrue (Ghosal et al. 2022. PLoS One 17:e0259238 contains over 1000 peer review documents, including sentiment analysis) and reflects a lack of review on the topic before examining this question.

I thank this reviewer for pointing me to this very useful study. I regret missing this one in my initial submission; I now discuss this paper in Pages 1 and 5 of the manuscript.

1. The author acknowledges the limitation that only papers under neuroscience were evaluated. Why not scale this method up to other fields within Nature Communications? Cross-field analysis of the features of interest would examine if these biases are present in other domains.

I share this reviewer’s opinion that it would be very interesting to expand this analysis to different subfields. I initially only included Neuroscience papers, because I was uncertain whether I would be able to properly assess the reviews from different scientific disciplines (and thus judge whether ChatGPT was able to provide plausible scores). The different reviewers have provide me with a list of potential follow-up experiments, and I am currently considering different options for future work, including expanding into different fields within Nature Communications. Additionally, I am looking to team up with a journal to perform the experiments laid out in (the new) Supplementary Fig. 6 of the new paper, to study whether I can find evidence of bias across rejected and accepted manuscript papers of a journal. I am also looking into ways to automate data collection using APIs, and by utilizing the rapidly expanding databases for transparent peer review. Yet, based on this preprint, I have received messages from academics that are interested in using generative AI to study scientific texts. By revising this manuscript now, I hope to provide them with the tools to concurrently expand the analysis of peer review into different scientific disciplines and journals.

The comments I received from the different reviewers made me realize that I did not describe the intent of this paper well enough in the original submission. I rewrote much of the Abstract, to emphasize the proof-of-concept nature of this study, and rewrote the Discussion to focus more on the limitations of the study.

**Reviewer #3 (Public review)**
Strengths:On the positive side, I thought the use of ChatGPT to score the sentiment of text was novel and interesting, and I was largely convinced by the parts of the methods which illustrate that the AI provides broadly similar sentiment and politeness scores to humans who were asked to rank a sub-set of the reviews. The paper is mostly clear and well-written, and tackles a question of importance and broad interest (i.e. the potential for bias in the peer review process, and the objectivity of peer review).

Thank you.

Weaknesses:The sample size and scope of the paper are a bit limited, and I have written a long list of recommendations/critiques covering diverse aspects including statistical/inferential issues, missing references, and suggestions for other material that could be included that would greatly increase the usefulness of the paper. A major limitation is that the paper focuses on published papers, and thus is a biased sample of all the reviews that were written, which prevents the paper properly answering the questions that it sets out to answer (e.g. is peer review repeatable, fair and objective).

I very much appreciate this reviewer taking the time to provide me with such a detailed list of recommendations. Below, I will respond to this list in a point-by-point manner.

**Reviewer #3 (Recommendations to author)**
My main issues with the paper are that it is not very ambitious, and gave me the impression the aim was to write the first paper using ChatGPT to address this question, rather than to conduct the most thorough and informative investigation that would have been feasible (many obvious questions that could be addressed are not tackled, since the sample size is small and restricted). There are also issues with selection bias, and the statistical analysis, that have possibly led to erroneous inferences and greatly limit what conclusions can be drawn from the analysis. I hope my comments of use in further improving the paper.The repeatability of ChatGPT when calculating the two linguistic characteristics is low. Taking the average of multiple assessments is one way to deal with this. To verify that taking the average of, say, 5 scores gives a repeatable score, the author could consider calculating 10 scores for a set of 20-30 reviews, calculating two scores for each review using the first 5 and second 5 ChatGPT ratings, and then calculating repeatability across the 20-30 reviews. It is important to demonstrate that ChatGPT is sufficiently repeatable for this new method to be useful.Also, it might be possible to automate this process a bit to save time - e.g. the author could change the ChatGPT prompt, like "please rate the politeness of this review from -100 to +100, do it 10 times independently, and print your 10 ratings as well as their average". Hopefully the AI is smart enough to provide 10 independently-computed ratings this way, saving the need to copypaste the prompt into the chat box 10 times per review.

This was a great recommendation by this reviewer, and a point also raised by Reviewer #1. Based on their suggestion, I looked into how each additional iteration of scoring would reduce the variability of scoring for a subset of papers (thus being able to advice users on an optimal number of iterations). I also tested this Reviewer’s suggestion to ask ChatGPT to score many times, and give separate scores for each iteration — this worked very well.

Interestingly, I observed that ChatGPT has become significantly more reliable in providing sentiment and politeness scores in recent versions. For the latest version (ChatGPT Aug 3, 2023), R2 = 0.992 for sentiment and R2 = 0.859 for politeness were reached for two subsequent iterations of scoring. Unfortunately, OpenAI does not allow access to previous version of ChatGPT, so the current dataset could not be re-scored. Yet, based on these data, there may no longer be a need for people to perform repeated scoring. I show these data in Supplementary Fig. 2, as I believe this is very useful information for people who are interested in using this tool.

To my mind, the main reason to use an AI instead of one or more human readers to rank the sentiment/politeness of peer reviews is to save time, and thereby allow this study to have a larger sample size than would be feasible using human readers. With this in mind, why did you choose to download only 200 papers, all from the discipline of Neuroscience, and only from Nature Communications? It seems like it would be relatively easy to download papers from many more journals, fields of research, or time periods if using AI-based methods, and in fact it would have been feasible (though fairly laborious) for one person to read and classify the sentiment of the reviews for 200 papers.As well as providing more precise estimates of the parameters you are interested in (e.g. the consistency of reviews, and the size of the difference in reviewer sentiment between author genders), expanding the sample beyond this small set of papers would allow you to address other interesting questions. For example, you could ask whether the patterns observed for neuroscience are similar to those in other research disciplines, whether Nature Comms is representative of all journals (given there are other journals with public reviews), and you could test whether the male-female differences have become greater or smaller over time (e.g. by comparing the male-female differences observed in the past to the effect size observed in 2022-23). Additionally, the main analyses in this paper would have higher statistical power - for example, you only include 53 papers with a female senior author, giving you quite low power/ precision to estimate the gender difference in the average sentiment of reviews (given the high variance in sentiment between papers).

I want to thank this reviewer for taking the time about possible ways to increase the impact of this work. I agree, these are all great suggestions, and there are many possibilities to apply ChatGPTbased natural language processing to scientific peer review. Respectfully, I chose to continue with publishing this work in the form of a proof-of-concept paper, because I currently do not have the resources to perform this (quite labor intensive) study. Below I will explain my reasoning, that I also shared with Reviewers #1 and #2.

I initially only included Neuroscience papers, because I was uncertain whether I would be able to properly assess the reviews from different scientific disciplines (and thus judge whether ChatGPT was able to provide plausible scores). The different reviewers have provide me with a list of potential follow-up experiments, and I am currently considering different options for future work, including expanding into different fields within Nature Communications. Additionally, I am looking to team up with a journal to perform the experiments laid out in (the new) Supplementary Fig. 6 of the new paper, to study whether I can find evidence of bias across rejected and accepted manuscript papers of a journal. I am also looking into ways to automate data collection using APIs, and by utilizing the rapidly expanding databases for transparent peer review. Yet, based on this preprint, I have received messages from academics that are interested in using generative AI to study scientific texts. By revising this manuscript now, I hope to provide them with the tools to concurrently expand the analysis of peer review into different scientific disciplines and journals. The comments I received from the different reviewers made me realize that I did not describe the intent of this paper well enough in the original submission. I rewrote much of the Abstract, to emphasize the proof-of-concept nature of this study, and rewrote the Discussion to focus more on the limitations of the study.

Also, if you could include some reviews of papers that were reviewed double-blind, you could test whether the gender-related differences in peer reviews are ameliorated by double-blind reviewing. Nature Comms (and many other journals with open review) do have some double-blinded papers, and there is evidence that that double-blinding is preferentially selected by authors who think they will experience discrimination in the peer review process (DOI: 10.1186/s41073-018-0049-z), and also that double-blinding does ameliorate bias (DOI: 10.1111/1365-2435.14259), so this seems very relevant to the ideas under study here.I note that the PLOS journals allow open peer review, and there is an API for PLOS which one can use to download the reviews for a given paper (e.g. try this query to get to the XML file of a paper which has open peer review: http://journals.plos.org/plosone/article/file?id=10.1371/ journal.pone.0239518&type=manuscript). Using an API could allow this project to be scaled up, because you can programmatically search for the papers with open reviews, download those reviews using the API and some code, and then score them using the same ChatGPT-based methods used for Nature Comms. Also, Publons recently merged with Web of Science (Clarivate), and you can now read all the open peer reviews on Web of Science for papers which had open review (e.g. for this paper: https://www-webofscience-com.napier.idm.oclc.org/wos/woscc/fullrecord/WOS:000615934800001). It would be possible to write to Web of Science, request access to their data or search engine, and programmatically download many thousands of papers and their associated reviews, and then use ChatGPT or a similar AI to score them all (especially if you can pass the reviews to ChatGPT for scoring programmatically, instead of manually copy-pasting the reviews into the chat box one at a time as it appears was done in the present study).

These are great suggestions, and I have different plans for follow-up studies, including the use of APIs to download large batches of peer reviews. The analyses in this paper have been performed in February of this year, even before the ChatGPT API had been released, which did not let me automate the process at that time. As a result, these analyses have been performed manually. I realize that the field is moving rapidly, and that there are now different options to scale this up quickly.

I plan on using the suggestions from this Reviewer for follow-up experiment in a next paper, and publish this revision as a proof-of-concept paper. In this way, different researchers can optimally use ChatGPT-based sentiment analyses for similar studies without a delay.

As you acknowledge, there is a selection bias in this study, since you only include papers that were ultimately published in Nature Comms (missing reviews of papers that were rejected). This is a really big limitation on the usefulness of some of your analyses. For example, you found no relationship between author institutional prestige and reviewer sentiment. This could be evidence of a fair and impartial review process (which seems unlikely!), or it could be a direct result of selection bias (specifically a "collider bias", like the famous example involving height and skill among professional basketball players). The likelihood that a paper is published is positively related both to its quality and the prestige held by the authors, we might expect a flatter (or even negative) correlation between prestige and reviewer sentiment among papers that were published than among the whole set of papers (like how the correlation between height and speed/skill is less positive among NBA players than among the general population, since both height and speed/skill provide advantages in basketball).

I agree with this reviewer that the selection bias is a major limitation of this study. I rewrote much of the Abstract and Discussion to tone down claims, and more prominently discuss the limitations of this study. I also made several suggestions for follow-up experiments.

In the section "Consistency across reviewers", you write that there was little similarity between review sentiment scores from different reviewers from the same paper, and then write "This surprising result indicates high levels of disagreement between the reviewers' favorability of a paper, suggesting that the peer review process is subjective." However I disagree with this conclusion for three reasons:Firstly, your dataset only includes papers that were published, and thus there is a selection bias against manuscripts where both/all reviewers disliked the paper - the removal of this (probably large) set of reviews will add a (potentially very strong) downward bias to your estimate of how consistent the review process is (since you are missing all those papers where the reviewers agreed). I think that one cannot properly answer the question "are reviewers consistent in their appraisals" without having access to papers that were rejected as well as those that were accepted.

I agree with this reviewer that there is a selection bias in this study, which I acknowledged throughout the initial submission of this manuscript. Indeed, having access to reviews of rejected papers will greatly increase my confidence in this finding. However, if there is consistency across reviewers in the entire pool of (post-review rejected+accepted) manuscripts, some of that has to trickle down into the pool of accepted papers. The correlation between sentiment scores of the different reviewers is so strikingly low (or even absent) that I simply cannot envision a way in which there is consistency across reviewers in the pre-editioral decision stage. Yet, I realize that this point is debatable. Therefore, I changed the phrasing of the Discussion section, including the following sentence:

That being said, the extremely low (or even absent) relation between how different reviewers scored the same paper was striking, at least to this author.

Secondly, the method used to assess whether the reviews for each paper tend to be similar (shown in Figure 3b) does not fully utilize the information contained in the data and could be replaced with another method. (In the paper 3 univariate regressions compare the sentiment scores for R1 vs R2, R1 vs R3, and R2 vs R3, which needlessly splits up the data in the case of papers with more than 2 reviewers, reducing power.) You could instead calculate the intraclass correlation coefficient (aka 'repeatability'), to determine what proportion of the variance in sentiment scores is between vs within papers (I suggest using the excellent R package rptR for this). Note that the sentiment scores are not normally distributed, and so regular regression (as you used) or one-way ANOVA (which you might be tempted to use for the ICC calculation) are not ideal - consider using a GLM or transformation (the rptR package automates the tricky calculation of repeatability for generalized models).

I thank this reviewer for pointing me towards this option. I added this analysis to Fig. 3b, which confirmed the inconsistency in sentiment scores for reviews of the same paper (ICC = 0.055). As suggested by this reviewer, I decided to perform the ICC on log-transformed data, as ICC calculation is very sensitive to non-normally distributed data.

Thirdly, an alternative and very plausible hypothesis for this lack of similarity (besides peer review being highly subjective) is that ChatGPT is estimating the "true sentiment" of a review (i.e. what the reviewer intended to say) with some amount of error (e.g. due to limitations/biases in the AI, or reviewers struggling to make themselves understood due to issues such as writing in a second language, typos, or writing under time pressure), which dilutes the similarly in the estimated sentiment of the reviews. In other words, if the true sentiment values are strongly correlated, but there is random error in how those values are estimated by ChatGPT, then the correlation between reviewer scores for each paper will tend to zero as the error tends to infinity. Furthermore a nebulous quality like "sentiment" cannot be fully summarised in a single variable running from -100 to +100, and if you had used a more multi-dimensional classification system for the reviews (or qualitative assessment by human readers) you might have found that there is a bit more correspondence (I'm speculating here, but I think you cannot really exclude this and the paper doesn't mention this limitation).

This point is well taken. I added caveats to the Discussion section on Page 5. Altogether, after taking these caveats into account, I do believe that this analysis convincingly demonstrates subjectivity in the peer review of this subset of papers. That said, I hope that my re-written discussion and additional analysis have added the necessary nuance to this point.

In Figure 3C, you write "Contribution of paper scores to review time". This strongly implies to the reader that the sentiment scores inferred for the reviews have a causal effect on the review time. This is imprecise writing (since the scores were calculated by you after the papers were published, and thus cannot be causal - you mean that the *actual reviews* affected the review time, not the scores), but more importantly you cannot infer any causality here since your dataset is observational/correlational. You could fix this by re-phrasing to emphasise this, e.g. "Statistical associations between paper scores and review time".

This is a very good point raised by this reviewer. I have corrected the phrasing so it no longer implies causality.

For the analysis shown in Figure 4d and Figure 4e, I am not certain what you mean by "data split per lowest/median/highest sentiment score". This is ambiguous, and I am also not sure what the purpose of this analysis is or what it shows - I suggest re-writing for greater clarity (and ideally providing the code used in all your analyses) and perhaps revising the analysis. Additionally, an important missing piece of information from this analysis (and most analyses in the paper) is the effect size. For example, you don't report what is the difference in politeness score and sentiment score between male and female authors, and what is the SE and 95% CIs for this difference. From eyeballing the figure, it looks like the difference in politeness is about 4 points on your 200point scale - this is small in absolute terms, but might be quite large in relative terms given that "politeness score" usually hovered around a small part of the full 200-point scale. What is this as a standardised effect size (i.e. in terms of standard deviations, as captured by effect sizes like Cohen's d and Hedges' g)? Calculating this (and its 95% CIs) would allow you to say whether the difference between genders is a "big effect", and give an idea of your confidence in your effect size estimate and any inferences drawn from it. You even discuss the effect size in your discussion, so it would help to calculate the standardised effect size. If you're not familiar with effect size and why it's useful, I found this paper very instructive: https://onlinelibrary.wiley.com/ doi/abs/10.1111/j.1469-185X.2007.00027.x

I agree with this reviewer that this phrasing was ambiguous. I now rephrased this on Page 4 of the manuscript:

To study whether these more impolite reviews for female first authors were due to an overall lower politeness score, or due to one or some of the reviewers being more impolite, I split the reviews for each paper by its lowest/median/highest politeness score. I observed that the lower politeness scores for first authors with a female name was driven by significantly lower low and median scores (Fig. 4d, bottom panel). Thus, the least polite reviews a paper received were even more impolite for papers with a female first author.

I also added effect sizes of the significant effects from Fig. 4 to its figure legend. Please note that the used statistical tests are non-parametric, so I reported the Hodges-Lehmann differences (which is the median of all possible pairwise differences between observations from the two groups).

"Double-blind peer review has been debated before, but has come under scrutiny for various reasons" - this is vague and unhelpful. I think it's worthwhile to properly engage with the debate and the substantial body of evidence in your paper, given your main focus is on potential bias in the review process based on authors' identities (e.g. gender, institutional prestige).

I thank the reviewer for pointing this out. I rephrased this sentence to indicate that there is evidence that it helps to remove certain forms of bias (Page 5):

To address this issue, double-blind peer review, where the authors' names are anonymized, could be implemented. Evidence suggests that this is useful in removing certain forms of bias from reviewing8,9, but has thus far not been widely implemented, perhaps because some studies have cast doubt on its merits21,22.

I have also added a Supplementary Fig. 6 to this paper, in which I lay out how my tool can be used to study bias by applying it to single- and double-blinded reviews (see also my answer to the other question about this topic below).

On a related note, in the first paragraph, when discussing the potential of single-blind review to allow reviewers to essentially discriminate against papers by women, there is a key missing citation. This year, the first truly experimental test of this hypothesis was published (DOI:10.1111/1365-2435.14259); a journal conducted a randomised controlled trial in which submitted manuscripts were reviewed either single- or double-blind. They found no effect of author gender on reviewer ratings or editorial decisions (though there was an effect of review type on success rate of authors from different countries). It would be better to cite this instead of reference 6, which as you acknowledge is methodologically flawed. This paper is also worth a read given your focus on Nature journals: DOI: 10.1186/s41073-018-0049-z.

This point is well taken. I now cite this paper (citation #8) and rephrased this part of theIntroduction (Page 1).

"Another - arguably more simple - solution [compared to double-blind peer review] could be for reviewers to be more mindful of their language use." Here, you seem to be saying that we don't need to blind author names during peer reviewers, because it would simpler if all reviewers were simply nicer! I object to this because (A) double-blind review is easy to implement, and greatly reduces the opportunity to tune the review to the author's identity (and there is some experimental evidence that it works in this regard), and (B) it seems like wishful thinking to say that we don't need to implement measures that reduce the scope for bias, because all reviewers could instead stop using impolite language.

This is a very valuable comment. I rephrased this to emphasize that this is an additional measure.

"reviewers may want to use ChatGPT to extract a politeness score for their review before submitting" Yes, that's an interesting idea, and I can imagine that some (probably small) proportion of reviewers will be interested in doing this. But I think you should think bigger about wholesale changes to the review system that are possible because of AI like ChatGPT. For example, the submission platforms where reviewers submit their reviewers (e.g. ScholarOne, Manuscript Central) could be updated to use AI to pre-screen draft reviews, and issue a warning to reviewers, like "Our AI assistant has indicated that the writing in this review might be impolite (*example phrases here*) - would you like to edit your review before you submit it?" Also, reviewcredit platforms like Publons could display not only the number of reviews that someone wrote, but an AI-generated assessment of how constructive, detailed, and polite their reviews are (this would help nudge people into writing better reviews, and also give credit where it's due to careful reviewers, which is part of the aim of Publons and similar platforms). This is just off the top of my head - there are many other good ideas about how AI could transform the peer review process. Indeed, AI is already good enough to generate quite useful peer reviews and constructive criticism of draft papers, and will surely get better at this... this surely has lots of implications for science publishing over the coming decades.

These are great suggestions for implementation of this tool. I now end the first paragraph of theDiscussion (Page 4) with the following sentence:

Such an automated language analysis of peer reviews can be used in different ways, such as afterthe-fact analyses (as has been done here), providing writing support for reviewers (for example by implementation in the journal submission portal), or by helping editors pick the best papers or most constructive reviewers.

"Further research is required to investigate the reasons behind this effect and to identify in what level of the academic system these differences emerge." Here you could mention what this research would be - I think you'd need the full sample of reviewed papers, not just those that were accepted. Spell out what analyses would be required to test and falsify the various (very plausible and interesting) competing hypotheses that you mention for the male-female difference in sentiment scores.

Great point. I added a Supplementary Fig. 6, in which I show a visual depiction of the experiments that can be performed to answer these questions.

"areas of concern were discovered within the academic publishing system that require immediate attention. One such area is the inconsistency between the reviews of the same paper, highlighting the need for greater standardization in the peer review process." I disagree here. I think it is natural for there to sometimes be differences in how two or more reviewers rate the quality of a paper, even if the peer review process were carefully standardised (e.g. via the use of a detailed "peer review form", which helps guide reviewers to comment on all important aspects of the paper - some journals use these). This is because reviewers differ in their experience, expertise, or interests, and so some reviewers will catch mistakes that others miss, or request stylistic changes that others would not. More broadly, it's often not possible to write a version of the paper that satisfies all possible reviewers.

I re-phrased part of the Discussion on Page 5 to indicate other sources of inter-reviewer variability. Specifically, I mention that some variability in sentiment can be expected based on the different backgrounds of the reviewers:

Notably, some level of variability may be expected, for example due to different backgrounds, experiences, and biases of the reviewers. In addition, ChatGPT may not always reliably assess a reviews sentiment, adding some spurious inter-reviewer variability.

Yet, as also mentioned in my response to one of the previous questions, I still find the the extremely low levels of consistency striking, even after taking these possible sources of interreviewer variability into account.

"the maximum score an institution could receive was 100 (in 2023 this was MassachusettsInstitute of Technology)" - this seems unnecessary information (just mention the score runs from 0-100).

I agree with this reviewer that this was unnecessary information. This has been removed.

"reviewers are generally familiar with the senior author of papers they review and thus are likely aware of their gender identity." This seems like a strong assumption, and you don't provide any evidence for it Speaking personally, as a reviewer and journal editor I am often not familiar with the senior author, or I *am* familiar with the first author - I am not sure how often I know the senior author but not the first author or vice versa. It's also not always the case that the first author is a junior scientist and the last author a senior, famous one, as you imply. I suggest that you use the same approach to score the gender of both author positions, namely inferring their gender programmatically from their name (I agree that generally the important thing for the purposes of this study is the gender that reviewers will infer from the name, not the author's actual gender, and so gender estimation from first names is the correct approach).

I appreciate this reviewer raising this point, and I have carefully weighed the pros and cons of both approaches. Initially, my intention was to rely only on a programmatic method to identify authors' names. However, I came to realize that there were inaccuracies in senior author gender predictions made by ChatGPT/Genderize. This was evident to me due to my personal familiarity with some of these authors, either because they are famous or through personal interactions. It seemed problematic to me to proceed with this analysis knowing that these misclassifications would introduce unnecessary variability to the dataset.

The advantage of the relatively small sample size in this study was the opportunity to manually perform this task, rather than being fully dependent on algorithms. While I attempted manual gender identification for the first author as well, this was way more challenging due to their limited online presence. The discrepancy in gender identification accuracy between first and senior authors did not go unnoticed, and I acknowledge the issue it presents. I also recognize that, unlike senior authors, reviewers may not necessarily be familiar with the first authors of the papers they evaluate, as indicated in the original submission of this paper. In light of this, I sought input from several PIs who often serve as reviewers. Their feedback confirmed that they typically possess knowledge of senior authors' identities, for example through conferences, whereas the same is not true for first authors. Yet, this may be different for other scientific disciplines, where the pool of reviewers might be bigger.

Notably, for future studies I may make a different decision, especially when I use larger datasets that require me to automate the process. I now more elaborately explain why I made this decision on Page 7 of the manuscript.

In the Abstract, you write "suggesting a gender disparity in academic publishing". This part of the sentence contains no information about what you think is the cause of the male/female difference, and no further interpretation of its ramifications, so I think you can just remove it (because "disparity" just means a difference, so you are effectively saying something redundant like "there was a difference between papers with male and female senior authors, suggesting there is a difference")

I thank the reviewer for pointing this out. I replaced the latter part of this sentence with “(…) for which I discuss potential causes.”, which I think is better than a short summary of potentialcauses which may lack the nuance that such a topic deserves.